# Comprehensive Spatio-Temporal Analysis of Travel Climate Comfort Degree and Rainstorm-Flood Disaster Risk in the China–Russia Border Region

**Yezhi Zhou** [1,2]**, Juanle Wang** [1,3,]*****, Elena Grigorieva** [4]**, Eugene Egidarev** [5]
**and Wenxuan Zhang** [1,2]

1   State Key Laboratory of Resources and Environmental Information System, Institute of Geographic Sciences
    and Natural Resources Research, Chinese Academy of Sciences, Beijing 100101, China;
    zhouyz@lreis.ac.cn (Y.Z.); zhangwx@lreis.ac.cn (W.Z.)
2   College of Geoscience and Surveying Engineering, China University of Mining & Technology (Beijing),
    Beijing 100083, China
3   Jiangsu Center for Collaborative Innovation in Geographical Information Resource Development and
    Application, Nanjing 210023, China
4   Institute for Complex Analysis of Regional Problems, Far-Eastern Branch of the Russian Academy of
    Sciences, 679016 Birobidzhan, Russia; eagrigor@yandex.ru
5   Pacific Geographical Institute, Far Eastern Branch of the Russian Academy of Sciences,
    690041 Vladivostok, Russia; egidarev@yandex.ru
*   Correspondence: wangjl@igsnrr.ac.cn; Tel.: +86-10-6488-8016

**Abstract:** Infrastructure and tourism is gradually increasing along the China–Russia border with the development of the China–Mongolia–Russia economic corridor. Facing the issues of thermal comfort and rainstorm-flood risk in the neighborhood area between China and Russia, we constructed homologous evaluation models to analyze spatial regularity and internal variations of their effect. Among the results, approximately 55% of the area was classified into the categories of "comfort" and "high comfort" in summer. Oppositely, the situation of most areas in winter corresponds to physical discomfort. On the other hand, the high-risk area of rainstorm-flood in spring and summer is principally located in the northern and southern regions, respectively, while this is further expanded in autumn. After that, the risk level turns to medium and low. Subsequently, a comprehensive assessment coordinate system of the two results was constructed to identify the distribution pattern of a seasonal suitable area for traveling in binary ways. The evaluation shows that Great Khingan Range in the north-western Heilongjiang province is the preferable place among most of seasons, especially in summer. While on the Russian side, the corresponding area is mainly spread over its southern coastal cities. The study is expected to provide recommendations for reasonable year-round travel time, space selection, and risk decision support for millions of people traveling between China and Russia.

**Keywords:** travel climate comfort degree; rainstorm-flood disaster risk; spatial-temporal assessment; combinatory analysis; China–Russia border

---

## 1. Introduction

In recent years, global climatic change has caused great variations in the degree of travel climate comfort and meteorological hazard risk in the human living environment [1]. Northeast China and the Russian Far East share a border of more than 3000 km. Under the development trend of the "the Belt and Road Initiative," the sound infrastructure in the important component of this area, Heilongjiang

province and Primorsky Krai, has made significant impact on the sustainable growth of the passenger flow [2]. According to statistical data, 353,000 Chinese citizens entered Primorsky Krai from January to September 2018, the main flow of tourism is connected with the presence of the sea and coast, at the same time, the proximity to the sea causes a large number of natural disasters. On the other hand, nearly half of the Russian tourists who travel to China choose to enter through the ports of Heilongjiang Province [3]. Therefore, considering the issues of the changing climate comfort situation with the frequent and severe meteorological disaster events, it is extremely relevant to help travelers perform activities at appropriate places and times.

Climate as a factor for tourism development has three important facets, and its thermal component is one of the most important [4]. Research on climate comfort assessment has a more than 50-year history and over 160 kinds of indices were applied into the work [5]. As early as 1966, Terjung suggested the concept of the climate comfort index [6] which attempted to integrate temperature, humidity, wind speed, and other meteorological parameters that affect physical and mental comfort into the comfort index (CI) and evaluate the physiological climate of the United States to discuss the feasibility of the evaluation scale. In 1968, this method was used to research the distribution pattern of global monthly climatic comfort [7]. In 1973, Oliver established the wind chill index table on the basis of the naked experiment [8] and in 1985, the Canadian meteorological bureau integrated the evaluation system of Terjung and Oliver to construct a standard model for the local climate comfort evaluation [9]. In the area investigated herein, Li et al. [10] calculated the comfort index of all parts of Heilongjiang province on the basis of the human comfort climate grade in Harbin and analyzed its annual changing characteristics and seasonal spatial distribution. Vitkina et al. and Veremchuk et al. [11,12] studied the impact of climate change on respiratory cases in Vladivostok from the perspective of climate and health and proposed corresponding countermeasures. Other scholars have tried to integrate human intervention factors into their research. For example, Ma et al. [13] used 30 popular tourism cities in China as research objects and built a spatiotemporal correlation model which was composed of the climate comfort evaluation results and tourists' network attention. Zhang studied the annual distribution differences of climate comfort in the Hubei province of China and analyzed its influence on the spatiotemporal evolution rule of passenger flow in the past 10 years [14]. These studies scientifically interpreted the relationship between climate comfort and human travel activities. Based on the above, the impact of climate comfort on human activities will always be a popular research topic.

While in terms of the regional characteristics of the study area, despite considering the climatic comfort affection, recognizing the influence of the disaster risk is also important [15,16], especially with the assessment of climate induced natural hazards, such as heavy rains and floods, etc., being the most important [4]. Therefore, data which related to both perspectives were used to depict the regionalization of their effect. The results are expected to provide referable guidance for safe and comfortable travel.

## 2. Materials and Methods

### 2.1. Study Area

This article studies Heilongjiang province in Northeast China and Primorsky Krai in the Russian Far East (Figure 1). The area lies between 118°53′ E–139°00′ E longitude and 38°43′ N–48°00′ N latitude and covers 637,672.2 km². According to the Köppen–Geiger climate classification map, Heilongjiang province has a temperate continental monsoon climate, and Primorsky Krai has a temperate marine monsoon climate [17]. The area has abundant annual precipitation between 400 and 800 mm in most places, approximately 80–90% of the total amount occur in the growing seasons while the annual variation is relatively stable. From the view of the drainage distribution, Heilongjiang province has three major basins: the Heilongjiang, Songhua, and Ussuri rivers, and there are 1918 rivers with an area of more than 50 square km. On the other side, more than 6000 rivers run in the region of Primorsky Krai, the longest of which is the Ussuri river basin (903 km), with more than 90 rivers over 50 km in

length. These natural geographical conditions cause rainstorms and floods to be the principal natural disasters in these areas. After counting the corresponding statistics which were obtained from the Emergency Events Database (https://www.emdat.be/database), it was found that the type of disaster event has occurred nine times in the study area from 2010 to 2017. In mid-August 2016, Heihe city in Heilongjiang province was hit by heavy rain that affected 79,000 people and 56,000 hectares of crops and caused direct economic losses of 94,447 million Chinese yuan [18]. Additionally, 10 municipal districts in Russia, including Ussuriysky and Khankaisky, were hit by the heaviest rainstorm relative to the rest of the districts of the same kind during 40 years, and more than 40,000 people were affected by the disaster, which directly caused economic losses of 2535 million Russian rubles [19].

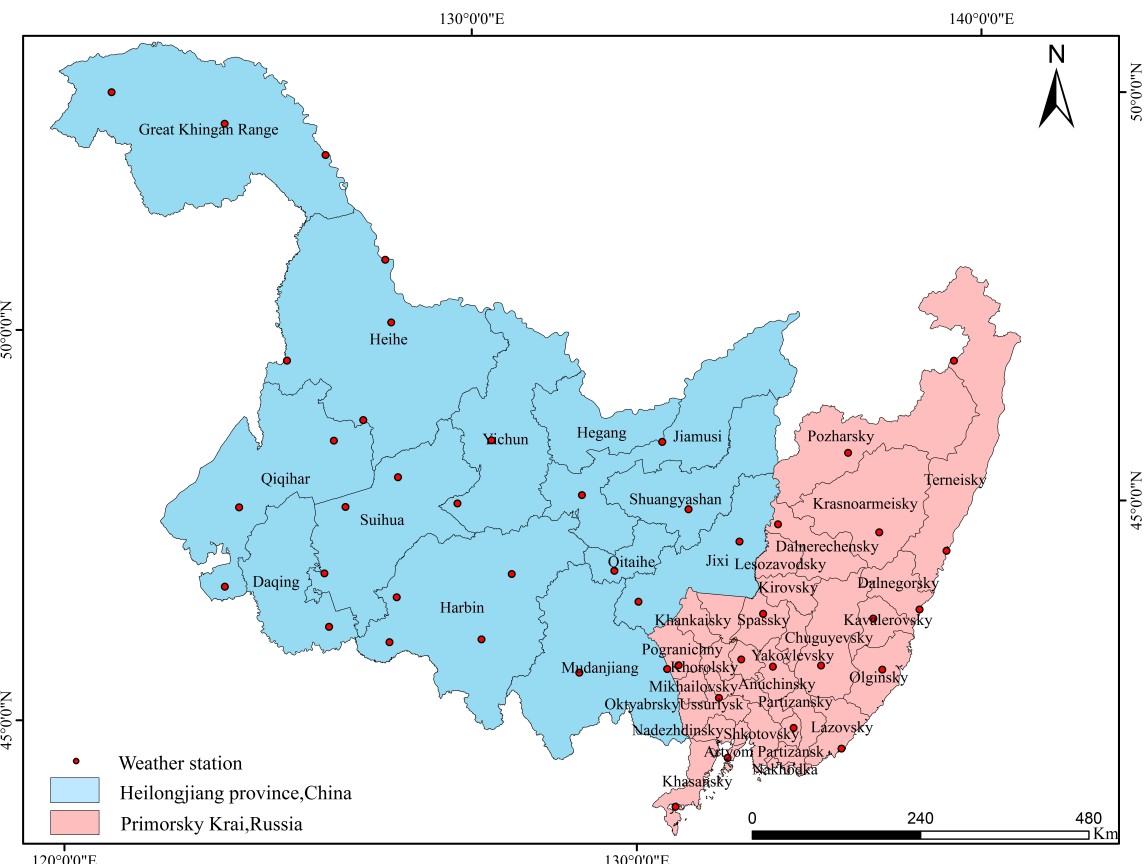

**Figure 1.** Geographical location and distribution of weather stations.

*2.2. Data Sources*

The data for the administrative divisions are obtained from the resource discipline innovation platform of the Chinese Academy of Sciences (http://www.data.ac.cn/). Meteorological data include four types of ground climate data such as air temperature, wind speed, relative humidity, and precipitation; they were collected for 46 weather stations from 1980 to 2016 (Figure 1). The Chinese meteorological data were obtained from the China meteorological science data sharing service website (http://cdc.cma.gov.cn/), and the related data of Primorsky Krai were obtained from the website of the All-Russian Research Institute of Hydrometeorological Information, World Data Center, in Obninsk (http://meteo.ru/). Digital elevation model (DEM) data with a spatial resolution of 90 m were obtained from the system of Shuttle Radar Topography Mission (SRTM), as jointly measured by National Aeronautics and Space Administration (NASA) and National Imagery and Mapping Agency (NIMA). Data of 1:200,000 water systems were obtained from Pacific Geographical Institute of the Far Eastern Branch of the Russian Academy of Sciences (http://tigdvo.ru/main.html). The data on land area, population, regional gross domestic product (GDP), and disaster censuses in China were from

the China Statistical Yearbook for Regional Economy and China Meteorological Disaster Yearbook. The corresponding data for the Russian side were from the federal state statistical office service website of Primorsky Krai (http://primstat.gks.ru/) and the website of Russian Federation emergency control ministry (https://mchs.gov.by/).

### *2.3. Methods*

#### 2.3.1. Evaluation Method for Travel Climate Comfort Degree (TCCD)

Travel climate comfort degree is a bio-meteorological index that measures the comfort state of the human body during their travel according to the environmental perception [20]. Three meteorological elements, air temperature, relative humidity, and wind speed combined with DEM are selected as the main evaluation elements. The combination of the above factors, indices of temperature-humidity index (THI) and wind chill index (WCI), are used to evaluate and analyze the comfort situation by referring to the corresponding somatosensory classification standards [21].

THI is determined by using a composition of temperature and humidity to estimate the thermal level. The physical significance of the model is that it improves the temperature evaluation index by accounting for the humidity [22]. The model is expressed as shown in Formula (1):

$$THI = t - 0.55(1 - 0.01RH)(t - 14.5) \tag{1}$$

WCI represents the effect of wind speed and air temperature on human heat dissipation in a cold environment. Physically, it represents the heat dissipation per unit area of body surface at a skin temperature of 33 °C [23]. The model is expressed as shown in Formula (2):

$$WCI = (33 - t)\left(9 + 10.9\sqrt{v}\right) - v \tag{2}$$

In the formulas above, $t$ denotes the air temperature (°C), $RH$ denotes the relative humidity (%), and $v$ denotes the wind speed (m/s).

The evaluation criteria based on the indices of THI and WCI are shown in Table 1.

**Table 1.** Somatosensory grade standards of the temperature-humidity index (THI) and wind chill index (WCI) [24].

| THI (F) Range of Value | Somatosensory Classification | WCI (kcal/m²·h) Range of Value |
|:---:|:---:|:---:|
| <40 | Extremely cold | <−1000 |
| 40–45 | Chilly | −1000−−800 |
| 45–55 | Cold | −800−−600 |
| 55–60 | Cool | −600−−300 |
| 60–65 | Clear and Cool | −300−−200 |
| 65–70 | Warm | −200−−50 |
| 70–75 | Tending toward hot | −50–80 |
| 75–80 | Hot | 80–160 |
| >80 | Sultry | >160 |

The data processing of all meteorological elements and the indices calculation of THI and WCI can be referred to in the relevant literature [25].

Referring to the operation scheme of previous literature [26], the evaluation model of travel climate comfort degree (TCCDEM) was constructed by Formula (3):

$$TCCDEM = 0.7 \cdot THI + 0.3 \cdot WCI, \tag{3}$$

where 0.7 and 0.3 are the weight coefficients of the THI and WCI, respectively.

The advanced evaluation criteria based on the evaluation model are shown in Table 2.

**Table 2.** Grade standards of the evaluation model of travel climate comfort degree (TCCDEM).

| TCCDEM Range of Value | Classification Level | Assignment of Critical Value |
|:---:|:---:|:---:|
| <−272 | Discomfort (e) | 4 |
| −272−−208.5 | Slight discomfort (d) | 3.25 |
| −208.5−−145 | Slight comfort (c) | 2.5 |
| −145−−81.5 | Comfort (b) | 1.75 |
| −81.5−−48 | High comfort (A) | 1 |
| −48−−14.5 | Comfort (B) | 1.75 |
| −14.5−64 | Slight comfort (C) | 2.5 |
| 64−104 | Slight discomfort (D) | 3.25 |
| >104 | Discomfort (E) | 4 |

### 2.3.2. Evaluation Method of Rainstorm-Flood Disaster Risk (RFDR)

Rainstorm-flood disaster risk refers to the possibility of the occurrence of the disaster events and their loss for human society [27]. This section provides a comprehensive analysis of the three disaster influencing indices based on meteorological, geomorphological, hydrological, and socio-economic data.

(1) Analysis of disaster-causing factor hazard (VE)

The index represents the severity of disaster-inducing factors in the assessment, which is principally determined by the intensity and frequency of regional precipitation. According to the regulations of the China Meteorological Administration, the standard for the occurrence of rainstorms is that the precipitation for at least one day reaches or exceeds 50 mm. The event ends after the first day without precipitation, and the precipitation of the whole process is accumulated.

The precipitation process sequence of the corresponding time duration was established according to the precipitation in the course of 1–10 days at each meteorological station by daily precipitation duration statistics (duration of more than 10 days were classified as 10-day conditions). Using the percentile method (Formula (4) to Formula (6)), the strength of the heavy rain events is divided into five levels, and the starting value of categories 1–5 corresponded to the 60%, 80%, 90%, 95%, and 98% percentile values of the sequence, respectively. Finally, all thresholds of the precipitation sequence are calculated as the critical precipitation of each grade.

$$p_i = (1 - \gamma) \cdot X_{(i)} + \gamma \cdot X_{(i+1)}, \tag{4}$$

$$j = \text{int}(p \cdot n + (1 + p)/3), \tag{5}$$

$$\gamma = p \cdot n + (1 + p)/(3 - j), \tag{6}$$

where $P_i$ denotes the $i^{\text{th}}$ percentile value, $X$ denotes the sample sequence in ascending order, $P$ denotes the percentile, $n$ denotes the total number of sequences, and $j$ denotes the $j^{\text{th}}$ sequence number.

According to the classification standard of critical disaster-causing precipitation, the occurrences of disaster processes in each grade were calculated, and the average results per 10 years were taken as the intensity frequency. Because the rainstorm strength grade is proportional to the level of disaster risk, the grade weights were designed for 1/15, 2/15, 3/15, 4/15, and 5/15. After normalizing the frequency data of all levels (Formula (7)), a weighted comprehensive evaluation method (Formula (8)) was used to obtain the disaster-causing factor hazard coefficient in each region and the natural breakpoint grading method, which can both reduce the differences of the same level and increase inter-level otherness, was used to classify the results.

$$D_{ij} = 0.5 + 0.5 \cdot \frac{A_{ij} - \min_i}{\max_i - \min_i}, \tag{7}$$

where $D_{ij}$ denotes the normalization number of the $i^{th}$ indicator in area $j$, $A_{ij}$ denotes the $i^{th}$ indicator in area $j$, and $min_i$ and $max_i$ denote the maximum and minimum values of the indicator, respectively.

$$V_j = \sum_{i=1}^{n}\left(W_i \cdot D_{ij}\right),\tag{8}$$

where $V_j$ denotes the total value of the evaluation factor, $W_i$ denotes the weight of $i^{th}$ indicator, $D_{ij}$ denotes the normalized value of the $i^{th}$ indicator for factor j, and $n$ denotes the number of evaluation indices.

(2) Analysis of disaster-inducing environmental sensitivity (VH)

The index reflects the sensitivity of the natural environment such as landforms and water distribution to disasters in the risk assessment. Among the above influencing factors, the comprehensive assignment standard of elevation and its standard deviation [28] can characterize the impact of the former factor on disaster risk. Elevation was obtained from DEM data, and the standard deviation of altitude was calculated using GIS software [29]. Specific criteria are shown in Table 3.

**Table 3.** Scoring criteria of terrain factors combined by altitude and its standard deviation.

| Altitude (m) | Standard Deviation of Altitude | | |
|---|---|---|---|
| | ≤1 | (1,10) | ≥10 |
| [0,100) | 0.9 | 0.8 | 0.7 |
| [100,300) | 0.8 | 0.7 | 0.6 |
| [300,700) | 0.7 | 0.6 | 0.5 |
| ≥700 | 0.6 | 0.5 | 0.4 |

The influence of water system factors on disaster risk can be reflected through the drainage density value which was calculated in GIS software by using stream data [30,31]. The specific calculation method can be found in the literature [32].

According to the field situation and the expert scoring method, the weights of topographic factors and river system factors on the impact of the disaster were determined to be 0.5 and 0.5, respectively. The corresponding data were normalized and weighted, and a comprehensive evaluation method was used to obtain the sensitivity index of the disaster-inducing environment. Finally, the natural breakpoint grading method was used to grade the results.

(3) Analysis of disaster-bearing body vulnerability (VS)

Disaster-bearing body vulnerability represents the actual loss caused by the disaster to human society in the risk assessment. The average population density (total population/land area) and economic barometer (regional GDP/land area) are selected as the evaluation factors. After the data normalization process, the expert scoring method is used to determine the weight of each factor 0.6 and 0.4, respectively. The annual vulnerability index of disaster-bearing bodies is then calculated using the weighted comprehensive analysis. The hierarchy of results was eventually classified by the natural breakpoint grading method.

After considering the ground condition and seeking expert opinions, the weight coefficients of each index are obtained to 0.5, 0.3 and 0.2, respectively, then the evaluation model of rainstorm-flood disaster risk (RFDREM) was constructed:

$$RFDREM = VE^{0.5} \cdot VH^{0.3} \cdot VS^{0.2}.\tag{9}$$

### 2.3.3. Combinatory Evaluation Method of the Travel Climatic Resources Based on the Coordinate System

In combination with the evaluation models on travel climate comfort and risk, a comprehensive assessment coordinate system of the two results was constructed to identify the distribution pattern of

a seasonal suitable area for traveling. For the concision and intuition of the evaluation, we reset the grade criteria of each model based on the former achievements.

(1) Reconstruction method of the grade criteria on travel climate comfort degree

Based on the former classification, the grades b, A, B ('comfort' to 'high comfort') are merged into one category (comfort) and the critical values of each classification level were assigned (Table 2). After the procession, the evaluation results were reclassified to 4 categories of quantitative expression as "comfort" (1–1.75), "Slight comfort"(1.75–2.5), "Slight discomfort" (2.5–3.25), "Discomfort" (3.25–4). Then the calculations of the evaluation model were mapped into the corresponding range by the new grade standards.

(2) Reconstruction method of the grade criteria on rainstorm-flood disaster risk

In the former study, the evaluation results were classified into 5 categories based on the risk intensity by the natural breakpoint grading method in GIS software. In this section, the percentile method was used to quantify the level. Firstly, the calculations of RFDREM in the four seasons were arranged in descending order, then the minimum and maximum values of the sequence (0.101 and 0.469) were selected as the critical value of low-risk and high-risk levels. Meanwhile, the calculation which located in the 1/3 and 2/3 position of the array (0.21 and 0.36) were extracted as the boundary indicators between low-medium risk and medium-high risk, respectively.

Finally, the coordinate system of comprehensive assessment was constructed. The values of its abscissa are the travel climate comfort degree (TCCD), which were calculated on the area of weather stations and the values of its ordinate are the corresponding calculations of rainstorm-flood disaster risk (RFDR). For readable analysis of the distribution, the scale of the values was magnified 100 times.

## 3. Results

### 3.1. Spatial and Temporal Dynamic Evaluation of Travel Climate Comfort Degree

The spatial calculation of the monthly travel climate comfort degree were performed by using the method described in 2.3.1, and four typical months (1, 4, 7, and 10) were selected to reflect the winter, spring, summer, and autumn evaluation results. The comfort levels of each season under the evaluation of TCCDEM were shown in Figure 2.

The most of area falls into the categories of physical discomfort in winter when evaluated with TCCDEM, including three levels of e, d, and c (Figure 2a). The range of grade "d" (chilly, slight discomfort) is the most widespread. Most of areas in Primorsky Krai and nearly half of Heilongjiang province are within this range. The only level out of the discomfort category— grade "c" (cold, slight comfort)—has the smallest comfort range, including only the Russian southeastern regions of Lazovsky and Olginsky. The high latitude area, which is located in northeast Heilongjiang province, is within the range of the "discomfort" class.

The travel climate comfort conditions gradually improve from north to south in spring (Figure 2b). The classification system under the evaluation model includes d, c, b, A and B. Among them, the range of grade "d" (Chilly, slight discomfort) is relatively small and includes only the edge of the mountain range in Great Khingan Range and Heihe city. The range of grade "c" (cold, slight comfort) is the largest and includes most areas in Heilongjiang province and the northern part of Primorsky Krai, accounting for nearly 60% of the total area. The grade "b" (cool, comfort) range is relatively large (13.8% proportion) and includes the cities in the southeastern part of Heilongjiang Province and the eastern part of the Russian side along the Pacific Ocean. Grade "A" (clear and cool, high comfort) and grade "B" (warm, comfort) are mainly distributed in the central mountainous area of Primorsky Krai, accounting for nearly 15% of the total area.

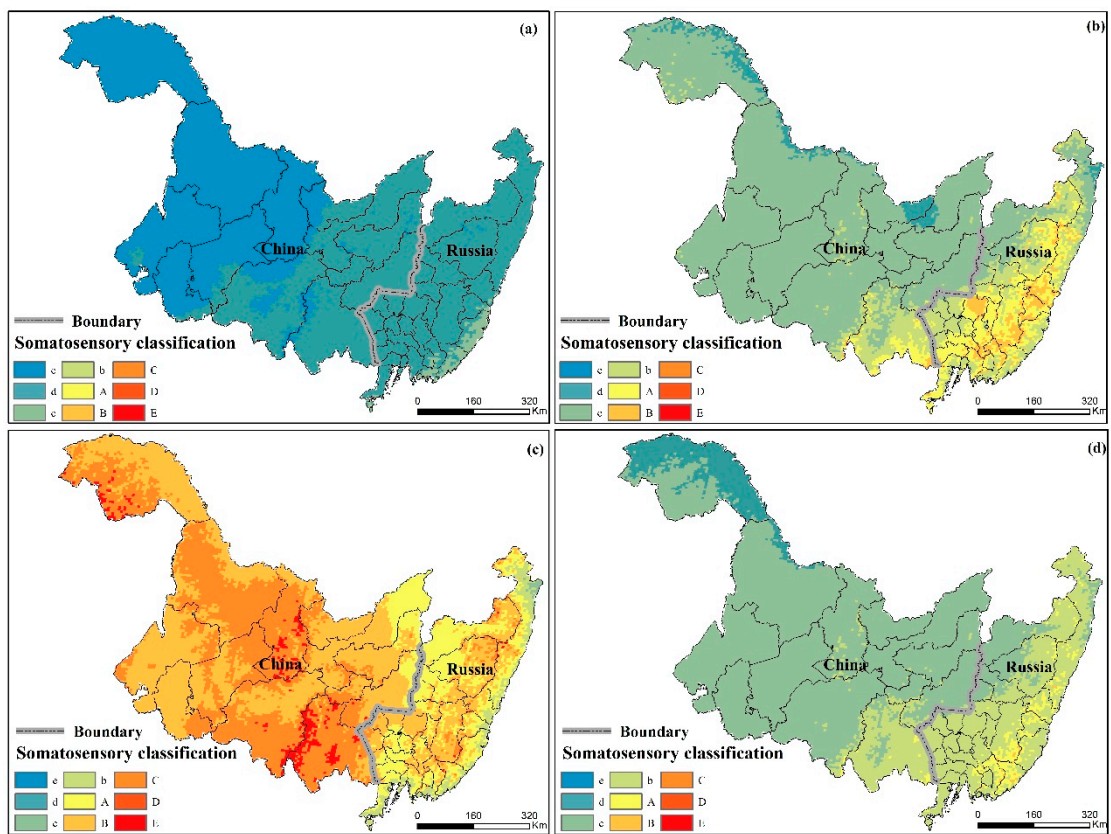

**Figure 2.** Regionalization of travel climate comfort degree in seasonal months. (**a**) Winter; (**b**) Spring; (**c**) Summer; (**d**) Autumn.

According to Figure 2c, the somatosensory classification of summer comfort conditions under the evaluation model includes six grades–c, b, A, B, C, and D. The ranges of grade "c" (cold, sight comfort) and grade "b" (cool, comfort) have a relatively low level of occupation and include only the northeastern end of Primorsky Krai. The ranges of grades "A" and "B" (comfort to high comfort) are relatively large (54.4% proportion); grade A is distributed in the northern China–Russia border area and in the eastern and southern coastal areas of Primorsky Krai, while grade B has a dispersive pattern which is scattered among the western and northeastern parts of Heilongjiang province and the central Russian side. Finally, the rest of Heilongjiang Province all belongs to the ranges of grade "C" and "D" (slight comfort to slight discomfort).

Figure 2d reveals that the classification system under the evaluation model includes four grades: d, c, b, and A. Among them, the range of grade "c" (cold, slight comfort) is the largest. Except for the north and south ends of Heilongjiang province, the remainder of the region combined with the northern part of Primorsky Krai belong to this range. The range of grade "b" (cool, comfort) is relatively large and includes the southern adjacent area and the eastern coastal zone of Primorsky Krai. Grade "A" (cool, highly comfortable) has the smallest range and includes the city of Partizansky and Chuguevskiy in Russia which is surrounded by the previous grade; the ratio of both ranges is nearly 30%. Finally, the rest Heilongjiang Province all belongs to the range of grade "d" (Chilly, slight discomfort).

*3.2. Spatial and Temporal Dynamic Evaluation of Rainstorm-Flood Disaster Risk*

In combination with the analysis methods of the three disaster evaluation indices above, spring extends from March to May; summer extends from June to August; autumn is set as September to November, and winter is set as December to February. The disaster risk regionalization under the evaluation of RFDREM in each season was conducted. The results were shown in Figure 3.

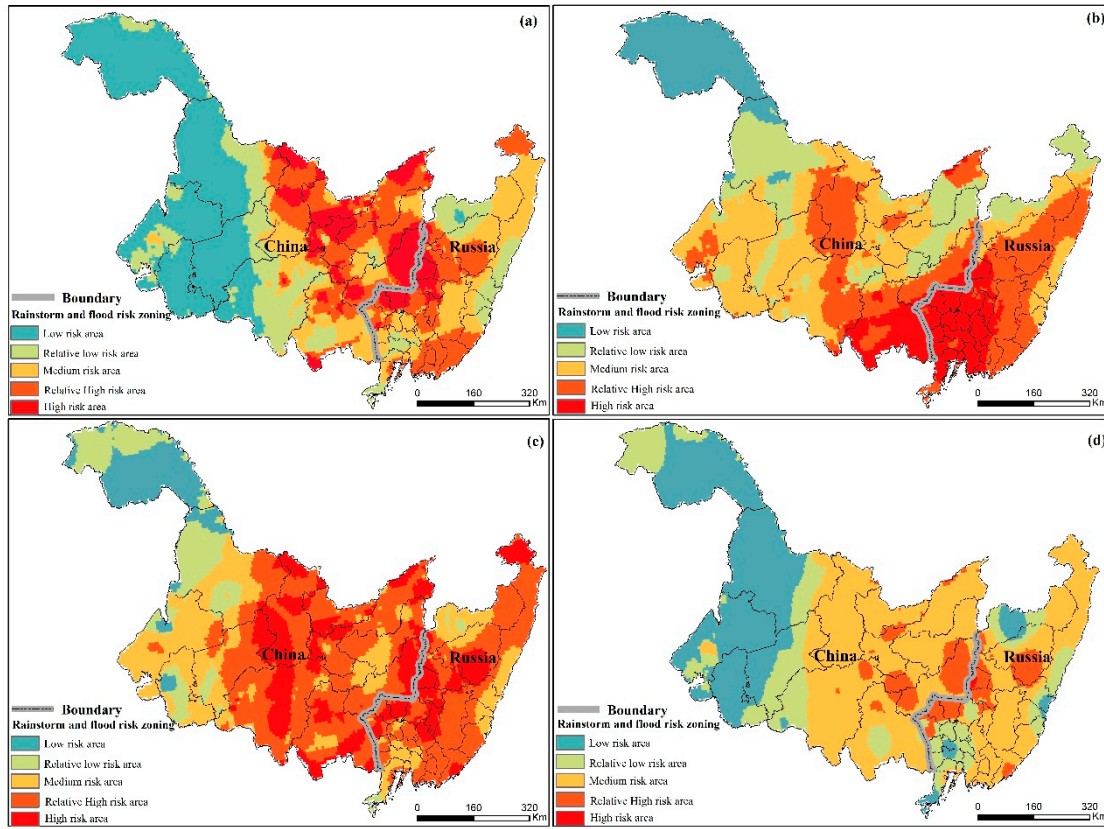

**Figure 3.** Seasonal regionalization of rainstorm-flood disaster risk. (**a**) Spring; (**b**) Summer; (**c**) Autumn; (**d**) Winter.

According to Figure 3a, the low-risk areas in spring are primarily located in the western cities of Heilongjiang province, and the areas in the south-central part of the province are primarily in the sub-low-risk and medium-risk areas; some areas in the northern part of Harbin are in a high-risk state. High-risk zones are primarily distributed in the urban agglomeration in the northeast of Heilongjiang province. On the Russian side, the disaster risk of western border cities increases sharply because of the influence of precipitation and the surrounding Ussuri river basin in the region, whereas in the south, municipal regions such as Lazovsky are in a high state of regional disaster risk because of the influence of the seasonal monsoon climate.

The condition of rainstorm-flood disaster risk shows an increasing trend from north to south in summer (Figure 3b). Among them, the Greater Khinggan range and some areas in Heihe city on the north side of Heilongjiang province have a relatively low disaster risk owing to the obvious topography changes. On the contrary, the areas in the southeastern orientation have a high disaster risk because of plentiful rainfall. Coincidentally, a similar risk condition is located in the southern city clusters of Primorsky Krai. The reason is perhaps attributed to the circumstance of relatively flat landforms and developed economical volume in this region. In some parts of the eastern coastal zone, such as the city of Kavalerovsk, the degree of rainstorm-flood disaster risk had slightly decreased to the medium level.

According to Figure 3c, the disaster risk situation generally presents a distribution trend of "lower in the east and west and higher in the center" in autumn (Figure 3c). As a similar condition as the former evaluation results, the corresponding disaster risk in the northwestern cities of Heilongjiang province is relatively low. Conversely, the northeastern part belongs to the Sanjiang Plain area, where densely-distributed rivers and explanate landscape cause a relatively high risk of disaster. On the Russian side, most of the cities are in a medium-high risk state. Among the target area, the city of Spassk Dalnii and its surrounding area, which is close to Khanka Lake on the Russian side, have the highest degree of disaster risk.

In winter, the distribution characteristic of the disaster risk is similar to that of autumn, while there is a general drop of one grade in most areas (Figure 3d). The western region of Heilongjiang province has the lowest risk level. Because of its natural geographical conditions, the environment of the central region is more sensitive, so the risk of disaster is further increased. The three evaluation indexes in the China–Russia border areas reached the highest level and indicate a relatively prominent risk for the region. In parts of Primorsky Krai, the disaster risk situation is relatively optimistic compared with other periods because the region has less precipitation in winter, and the population density and socio-economic development situation have relatively faded. Therefore, the risk of disaster is classified as the level of medium to low.

### 3.3. Comprehensive Evaluation of Travel Climate Comfort Degree and Rainstorm-Flood Disaster Risk

The evaluation results of each weather station based on TCCDEM and RFDREM were regraded according to Table 4, and the travel climatic resource conditions were analyzed and evaluated in multiple ways. The outcomes are shown in Figure 4.

**Table 4.** Grade criteria of climatic comfort and risk in a comprehensive assessment system.

| Range of Travel Climate Comfort Degree Classification Threshold | Category | Range of Rainstorm-Flood Disaster Risk Classification Threshold | Category |
|---|---|---|---|
| 100–175 | Comfort | 10–21 | Low risk |
| 175–250 | Slight comfort | 21–36 | Medium risk |
| 250–325 | Slight discomfort | 36–47 | High risk |
| ≥325 | Discomfort | | |

The travel climatic assessment results of spring, summer, and autumn are mainly located in the first half of the coordinate system (Figure 4). In summer, the two cities of the Greater Khinggan Range and Heihe in the northwest of Heilongjiang province locate in the comfort-low risk zone, whereas the climate comfort of this area is somewhat weakened in spring and autumn, while the risk degree remains stable. As the both ranges of comfort-medium risk and comfort-high risk, most of located areas are mainly distributed in Russian border cities such as Pogranichny and Khankaysky in spring. During the period of summer and autumn, the distribution of the target area turns to the direction of the northwestern Heilongjiang Province and the Russian eastern part instead. For the distribution pattern of slight comfort-medium and high risk plates, most of the areas which fall into those zones in summer and autumn are the coastal cities in southern Primorsky Krai, the evaluation results of Khasansky district in both seasons are within the range among the distributions. The climatic assessment results of winter travel are mostly located in another part of the coordinate system which means the travel climate comfort level is in the status of physical discomfort. Meanwhile, risk distribution during this period shows strong regional regularity. Jiamusi, Hegang, and other northeastern regions in Heilongjiang province have the highest risk degree, followed by western urban areas such as Qiqihar and Daqing, whereas the disaster risk along the northwestern to southeastern routes of the province is the lowest. In Primorsky Krai, the border cities along the Ussuri river basin have the highest disaster risk. The eastern and southern parts of the region are less prone to disaster occurrence. The disaster risk in other areas remains at a medium level.

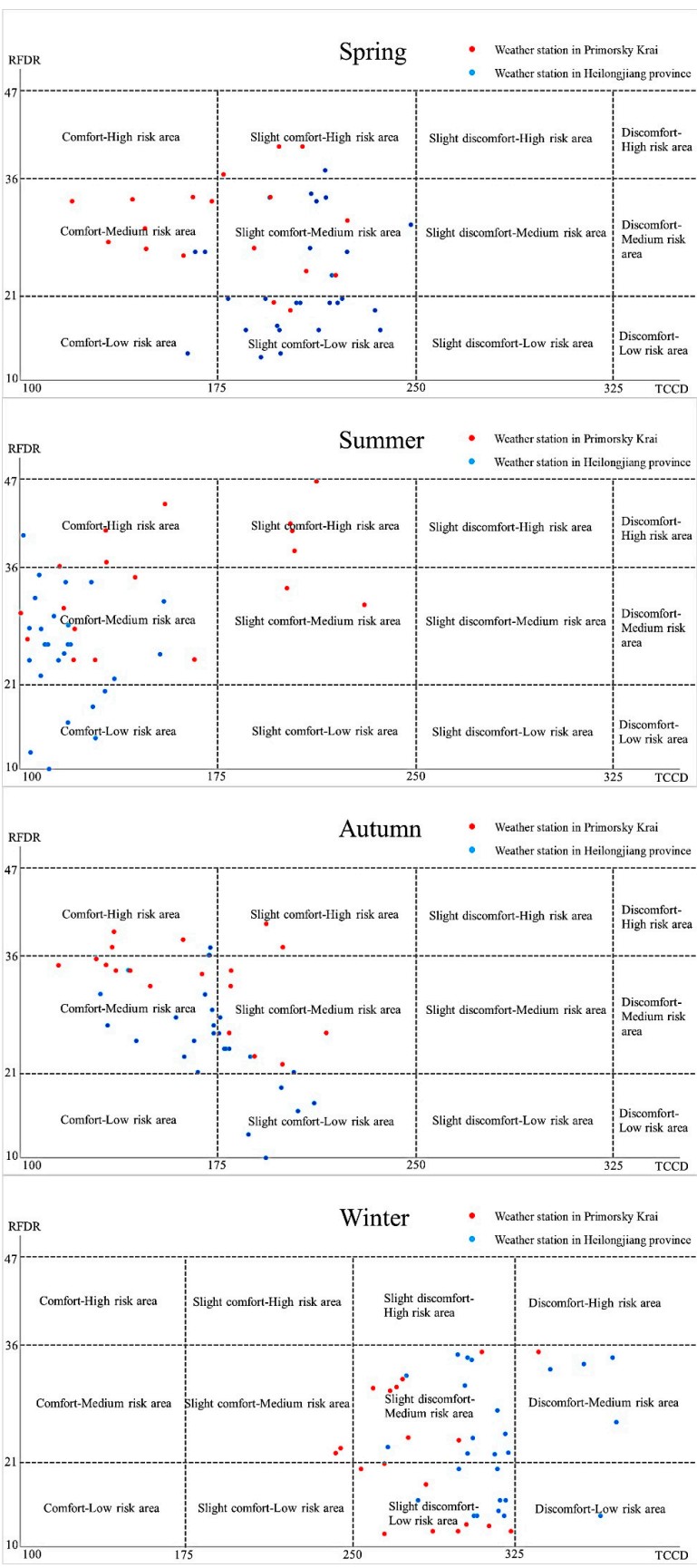

**Figure 4.** The combined analysis of climate comfort and flood disaster risk in the four seasons.

## 4. Discussion

From a seasonal perspective, the climate comfort evaluation results generally show that summer has the most comfortable climate areas. Spring and autumn are second, and the spatial distribution pattern of the comfort level in each region is similar. The comfort conditions in winter are the worst, and there is no climatic comfort zone under the corresponding evaluation model used in this study. Considering the climatic features of each season, temperature factor may intuitively play a leading role in the changing results under the evaluation model and this inference was echoed in other relevant studies [33]. From a regional perspective, the climate comfort situation in Russia is generally better than that in China in all seasons. Additionally, the eastern inner region of the Russian side is better than the western part, and the southern part is better than the northern part, except in summer. Those findings are perhaps due to the fact that the coastal areas in eastern Primorsky Krai are affected by the maritime climate influence throughout the year. Because of existing high heat capacity and the thermal conductivity of physical properties [34], the ocean heats the coastal area more quickly than inland areas in spring and slows the cooling process in autumn and winter. During the summer, a large amount of evaporation in the ocean can effectively improve the high thermal environment on land, so as to provide a more moderate and humid somatosomatic state for travelers. In spring, autumn and winter, because of the inverse relationship between latitude and temperature, the proportion of the area with cold and uncomfortable climatic conditions is relatively small in the southern Primorye region.

The risk assessment of rainstorm-flood disasters in each quarter shows the following distribution characteristics: the high-risk areas in autumn have the highest proportion (over 40%) of the disaster level conditions. The existing areas are concentrated in the central part of Heilongjiang province and the areas on both sides of the border between China and Russia. The results are basically consistent with the distributions of the affected population and direct economic losses recorded by the official statistics. The proportion of high-risk disaster areas in spring and summer are similar while the corresponding core areas are located in different directions (northern and southern part, respectively). The reasons behind these circumstances can be explained by the gradually warming weather conditions in spring which lead to a large amount of snow melt flowing into thawing river channels, which distribute densely in the northern part, and the circulation of warm and cold air mass make this region prone to large scale rainfall events. In summer, the Sea of Japan in the southern part was frequently hit by offshore typhoons [35]. Disasters often occur with super-heavy rainfall and a sharp increase in water level. Both of these influences are responsible for the sharp increase in the high-risk area. Precipitation is low in winter because of the influence of temperature, and some people in the area migrate to avoid the cold. Therefore, the condition of disaster risk in this season is generally optimistic. However, because of the impact of climate warming in recent years, the potential risk of disasters in river-dense areas is still not negligible. Local departments should always monitor river basin dynamics against the early occurrence of spring flood events.

Combined with the comprehensive evaluation results of TCCD and RFDR, it can be found that the Greater Khingan range and Heihe are the best travel choices for tourists in summer perhaps due to the geographical conditions of high latitude and altitude which not only enhance the somatosensory comfort level under the hot and humid environment, but weaken the capacity of rainfall accumulation as well. Additionally, the area is sparsely populated and its main economic activity is forestry. Previous studies proved that the water storage capacity of vegetation has a certain retardation effect on flood disasters [36]. Therefore, based on the advantageous geographical location and climate resources, the local government is advised to energetically develop mountain tourism such as hiking, climbing and other leisure activities including understory picking or agritainment. Considering the condition of travel climate resources among the Russian side, the risk of rainstorms and floods is generally high, especially for the capital city (Vladivostok) and its surrounding areas. While according to the results of TCCD, the region has a good climate comfort situation in summer and autumn. Based on the above characteristics, the recommendation to local government is to visualize the seasonal dynamic climate comfort degree map and rainstorm-flood disaster risk assessment results in the

areas of tourist attractions. For the travelers, getting relevant information in time is essential during their journey. According to the ornamental value of marine sightseeing and the richness of fishery resources in this region, the local government should take appropriate measures such as improving the recreational facilities for developing a cruise tour and constructing the artificial reef to provide a halobios reproduction site with a consideration of the current condition of climatic comfort and risk [37]. The aim is to achieve the mutually beneficial goal which allows travelers to enjoy the physical and mental pleasure brought by natural climate conditions and also promotes the economic development of related industries in the China–Russia border area.

Compared with TCCDEM and RFDREM, the corresponding results under the comprehensive assessment coordinate system were acquired in a similar way, either by the overall trend or by the details of distribution. So, the evaluation was, to some degree, a reference for helping travelers to recognize the climatic resource situation in the China–Russia border region from multiple points of view. However, there still exist a few distinctions between the former and latter results owing to the grade reconstruction. Some advanced means which can enhance both coherence of appraisals and intuition of analysis will be considered to utilize in the further study. In the evaluation of the coordinate system, the TCCD and RFDR values obtained from the weather stations were adopted to represent the corresponding conditions of its region. Additionally, due to insufficient data, this paper does not take disaster resistance and mitigation ability as one of the evaluation factors in the RFDREM, so the quantitative characterization of this aspect is not consummate. For the above imperfections, data which both provide successive information and express the capacity of disaster resistance will be collected to further improve the capacity of integrated evaluation in this transboundary region.

## 5. Conclusions

Heilongjiang province and Primorsky Krai, areas in a border region between China and Russia, were studied to analyze the spatiotemporal dynamic changes of TCCD and RFDR. Then, the regional distribution of suitable locations for travel in each season was comprehensively evaluated using a two-dimensional evaluation coordinate system. The results show that: (1) climate comfort evaluation in summer shows an affirmative result, with approximately 55% of the whole area having a physical level of "comfort" to "high comfort", whereas the corresponding area in autumn and spring accounts for approximately 30% and 28.5%, respectively; the comfort situation in winter is unsatisfactory, and the most area belongs to the category of physical discomfort; (2) risk of rainstorm-flood distributed unevenly both in spatial and temporal. In spring, the high risk areas are primarily concentrated in the northeast of Heilongjiang province and account for 26.9% of the total region. In summer, the corresponding area increased to 30%, and the core area moved to the south. In autumn, the corresponding area further expanded to the north, accounting for about half of the research area. In winter, the disaster risk was relatively reduced, and the medium and low risk became the mainstream of the intensity. (3) Most cities in summer have comfortable conditions, but the high possibility of disaster occurrences should not be ignored. The distribution of suitable travel areas in spring and autumn are mainly characterized by general comfort with medium and low risk, but the distribution is slightly different. There is no objective meteorological condition suitable for travel in winter; therefore, vulnerable tourists should reduce travel activities in the area appropriately according to individual conditions.

**Author Contributions:** J.W. was responsible for the research design, analysis, and designed and reviewed the manuscript. Y.Z. drafted the manuscript and was responsible for data preparation, experiments and analyses. E.E., E.G. and W.Z. participated in data collection and processing. J.W., Y.Z., E.G. and E.E. organized and participated in the field work of the study. All authors contributed to the editing and reviewing of the manuscript. All authors have read and agreed to the published version of the manuscript.

**Funding:** This research was funded by the Strategic Priority Research Program (Class A) of the Chinese Academy of Sciences grant number XDA20030203012, Special Exchange Program of Chinese Academy of Sciences grant number Y9X90050Y2, and the Construction Project of the China Knowledge Center for Engineering Sciences and Technology grant number CKCEST-2019–3-6.

**Conflicts of Interest:** The authors declare no conflict of interest.

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
