# Peer review of "Comprehensive Spatio-Temporal Analysis of Travel Climate Comfort Degree and Rainstorm-Flood Disaster Risk in the China–Russia Border Region"

_sustainability, doi:10.3390/su12083254_

Round 1

Reviewer 1 Report

This research constructed homologous evaluation models to analysis spatial regularity and internal variations of the effect on the issues of thermal comfort and rainstorm-flood-risk in the neighbourhood area, between China and Russia.  Researchers used meteorological data include four types of ground climate data such as temperature, wind, speed, relative humidity, and precipitation which were collected for 46 weather stations from 1980 to 2016. The research study aims to provide recommendation for reasonable year-round travel time, space selection, and risk decision support for millions of people travelling between China and Russia as the infrastructure and tourism are gradually increasing along the China-Russia border with the development of China-Mongolia-Russia economic corridor.

As the research results are expected to provide referable guidance for the safe and comfortable travel along the China-Russia border with the development of China-Mongolia-Russia economic corridor, authors are suggested to make the following discussion for adding value to the tourism industry and practitioners:

Researchers stated that risk of rainstorm-flood distributed unevenly both in spatial and temporal, and most cities in summer have comfortable conditions, but the high possibility of disaster occurrences should not be ignored; so, what sort of prevention or safety measures will be advised for the local government and tourism industry to follow?

Researchers commented that local government should take appropriate measures to intervene in the potential risk to ensure the safety of travellers and the smooth flow of the economic trade between China and Russia.  It is suggested that researchers should further propose what sort of appropriate measures should be taken for the ornamental value of the marine resource and the richness of fishery resources along the China-Russia border?

Researchers suggested that the local government should “make good use of the advantages of climate resources” to provide tourists with more comfortable and safe travel based on conditions stated in the paper.  It would be valuable for researchers to propose some concrete suggestions to utilise the climate resource advantageous for developing the tourism business in the neighbourhood area between China and Russia.

Author Response

Dear esteemed reviewer,

Thanks very much for these helpful comments!

Please see the response in the attachment.

Reviewer 2 Report

The main idea of the paper:
Study of the spatiotemporal dynamic changes of TCCD and RFDR of a border region between Russia and China, provides recommendation on travel and advisories.

1. Originality and significance of the research
Original to my knowledge

2. Technical and theoretical correctness
Very sound and theoretically correct

3. Readability of the paper
Readable and English is good.

4. Evaluation result
Discussion thorough and well done, evaluation is understandable to laymen.

5. Scope of the work.
Valid for the field of study of this topic.

Author Response

Dear esteemed reviewer

Thanks very much for the positive and encouraging evaluation for our manuscript.

Reviewer 3 Report

Dear Authors,

The article has a very interesting subject matter. It is a reference point, helping travelers to recognize the situation of climate resources on the border between China and Russia. The authors have precisely defined the risk of unforeseen situations depending on the season. They also referred to the probability of a natural disaster. Methodological issues do not raise objections. I believe that the article is an important source of knowledge. The data is presented in a clear, precise manner.

Author Response

Dear esteemed reviewer,

Thanks very much for the positive and encouraging evaluation for our manuscript.

The presentations of the discussions and results are improved by the revision of our research team.